# SAGEPHOS: SAGE BIO-COUPLED AND AUGMENTED FUSION FOR PHOSPHORYLATION SITE DETECTION

**Jingjie Zhang[1], Hanqun Cao[1], Zijun Gao[1], Xiaorui Wang[2], Chunbin Gu[1]***

[1]Department of Computer Science and Engineering, The Chinese University of Hong Kong
[2]College of Pharmaceutical Sciences, Zhejiang University
{jjzhang24, hanquncao2001, 1155224009}@link.cuhk.edu.hk,
wangxr2018@lzu.edu.cn, cbgu@cuhk.edu.hk

## ABSTRACT

Phosphorylation site prediction based on kinase-substrate interaction plays a vital role in understanding cellular signaling pathways and disease mechanisms. Computational methods for this task can be categorized into kinase-family-focused and individual kinase-targeted approaches. Individual kinase-targeted methods have gained prominence for their ability to explore a broader protein space and provide more precise target information for kinase inhibitors. However, most existing individual kinase-based approaches focus solely on sequence inputs, neglecting crucial structural information. To address this limitation, we introduce SAGEPhos (Structure-aware kinAse-substrate bio-coupled and bio-auGmented nEtwork for Phosphorylation site prediction), a novel framework that modifies the semantic space of main protein inputs using auxiliary inputs at two distinct modality levels. At the inter-modality level, SAGEPhos introduces a Bio-Coupled Modal Fusion method, distilling essential kinase sequence information to refine task-oriented local substrate feature space, creating a shared semantic space that captures crucial kinase-substrate interaction patterns. Within the substrate's intra-modality domain, it focuses on Bio-Augmented Fusion, emphasizing 2D local sequence information while selectively incorporating 3D spatial information from predicted structures to complement the sequence space. Moreover, to address the lack of structural information in current datasets, we contribute a new, refined phosphorylation site prediction dataset, which incorporates crucial structural elements and will serve as a new benchmark for the field. Experimental results demonstrate that SAGEPhos significantly outperforms baseline methods, notably achieving almost 10% and 12% improvements in prediction accuracy and AUC-ROC, respectively. We further demonstrate our algorithm's robustness and generalization through stable results across varied data partitions and significant improvements in zero-shot scenarios. These results underscore the effectiveness of constructing a larger and more precise protein space in advancing the state-of-the-art in phosphorylation site prediction. We release the SAGEPhos models and code at https://github.com/ZhangJJ26/SAGEPhos.

## 1 INTRODUCTION

Post-translational modifications (PTMs) are chemical alterations that occur to proteins after their initial synthesis by the cell's protein-making machinery. These modifications serve as crucial refinement mechanisms (Parekh & Rohlff, 1997; Xu & Chou, 2016), capable of altering a protein's function, localization, or interactions with other molecules. Phosphorylation, one of the most significant PTMs, allowing cells to regulate a diverse array of processes including metabolic pathways and kinase cascade activation, by activating or deactivating them. At the molecular level, as depicted in Fig.1 and catalyzed by protein kinases (Manning et al., 2002), phosphorylation is the process of transferring a phosphate group from a donor molecule, typically adenosine triphosphate (ATP), to a target protein, known as the substrate. This transfer primarily targets serine (S), threonine (T), or

---

*Corresponding Author.

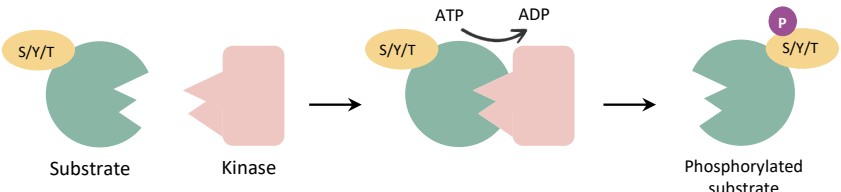

Figure 1: The process of phosphorylation. A kinase (pink rectangle) catalyzes the transfer of a phosphate group ("P") from ATP to a substrate (green circle), creating a phosphorylated substrate.

tyrosine (Y) residues on the substrate. ATP, the donor molecule, is a high-energy compound often described as the "energy currency" of the cell, due to its central role in fueling cellular activities. Upon donating a phosphate group during the phosphorylation process, ATP is converted into adenosine diphosphate (ADP), a molecule with two phosphate groups. The attachment of a phosphate group to a protein can potentially alter its function or activity, leading to perturbations in cellular homeostasis and signaling cascades. These disruptions have been linked to various human diseases, including cancer, neurodegenerative disorders such as Alzheimer's and Parkinson's diseases, and heart disease (Viatour et al., 2005; Nsiah-Sefaa & McKenzie, 2016). Therefore, the ability to accurately predict phosphorylation sites is crucial for understanding complex cellular networks and developing targeted therapies for various diseases.

Early phosphorylation site prediction methods, such as those used by Scansite 2.0 (Obenauer et al., 2003), DISPHOS (Iakoucheva et al., 2004), Musite (Gao et al., 2010), and RF-Phos (Ismail et al., 2016), relied on traditional techniques like position-specific scoring matrices and machine learning algrithoms such as random forest, but were limited by small datasets and the complexity of phosphorylation sites.

The advent of high-throughput mass spectrometry techniques (Salzano & Crescenzi, 2005) marked a significant turning point, leading to a dramatic increase in the discovery of phosphorylation sites. This technological advancement resulted in the identification of over 200,000 sites in the human proteome alone (Hornbeck et al., 2012). While this wealth of data presented unprecedented opportunities, it also introduced new challenges, particularly in terms of data quality, false positive rates, and the ability to process and interpret such large-scale datasets effectively.

To address these challenges, the field has increasingly turned to artificial intelligence techniques, particularly deep learning models. Recent approaches such as DeepPhos (Luo et al., 2019), MusiteDeep (Wang et al., 2017), and PhoSIDN (Yang et al., 2021) have shown improved accuracy by leveraging large datasets to identify complex patterns. However, many of these models rely solely on substrate sequences or incorporate only well-studied kinases, lacking the ability to comprehensively annotate specific kinases for each phosphorylation site (Needham et al., 2019).

A significant advancement came with Phosformer (Zhou et al., 2023), which advanced the field with a Transformer-based approach that adds a separate kinase sequence annotation for each phosphorylation site, demonstrating promising results. Despite its advances, Phosformer's limitations include a smaller pre-training dataset compared to Evolutionary Scale Modeling 2 (ESM2) (Rives et al., 2021; Lin et al., 2023) and a lack of Multiple Sequence Alignment (MSA) use, missing evolutionary context. Additionally, it does not integrate structural information crucial for accurate prediction (Iakoucheva et al., 2004).

To address existing challenges, we present SAGEPhos, a Structure-aware kinAse-substrate bio-coupled and bio-auGmented nEtwork for Phosphorylation site prediction. This novel model integrates inter- and intra-modality information, leveraging ESM2 for evolutionary context-aware sequence processing and combining multi-modal inputs through gated and residual networks. The gated network preserves comprehensive primary modality information, while the residual network selectively incorporates relevant secondary modality feature. We dynamically weighted the two networks, enabling the secondary modality data to modify the semantic space of main protein inputs, forming functional fusion patterns we term "Bio-Coupled" and "Bio-Augmented". In the inter-modal dimension of substrate and kinase analysis, we employ "Bio-Coupled" Fusion. Given the larger volume of task-relevant local substrate data versus the smaller number of kinase sequences,

we prioritize substrate data as our primary information source. This Bio-Coupled Fusion strategy mitigates noise from kinase sequences while allowing kinase information to modify the substrate's semantic space, creating a shared representation that captures essential kinase-substrate interaction patterns. For intra-modal analysis of substrate local sequence and structure, we implement "Bio-Augmented" Fusion. In this context, sequence data is accurate but lacks structural information, while structural details are predicted using sequence but potentially noisy. We therefore prioritize sequence as the primary modality, supplemented by structural data to enrich the representation of the substrate's local environment. Incorporating these insights, SAGEPhos is designed to extract and utilize critical information from both modalities, enhancing primary data with targeted secondary details while minimizing extraneous noise. This approach ensures high precision, offering advanced insights into cellular signaling and disease mechanisms. Experimental results across diverse datasets demonstrate SAGEPhos's superior performance in phosphorylation site prediction tasks, highlighting its ability to capture effective patterns.

The main contributions of this study can be succinctly summarized as follows:

(1) Our approach enhances phosphorylation site prediction by incorporating substrate 3D structural information, a key advancement over previous methods. We innovatively introduce modal priors into feature fusion, proposing a novel concept that treats the substrate as the primary modality and the kinase as an auxiliary one.

(2) SAGEPhos employs "Bio-Coupled" Fusion for inter-modality analysis, utilizing kinase data to refine substrate space, capture crucial interaction patterns in a shared semantic space. For intra-modality, it uses "Bio-Augmented" Fusion, emphasizing sequence data and judiciously integrating structural details to enhance substrate representation.

(3) Comprehensive experiments demonstrate that SAGEPhos accurately captures kinase-substrate interactions, local sequence contexts, and spatial accessibility of potential sites, demonstrating superior performance on diverse benchmark datasets.

## 2 RELATED WORK

**Phosphorylation Site Prediction.** Phosphorylation plays a crucial role in numerous biological functions (Huang et al., 2016). The domain of phosphorylation site prediction has significantly progressed, with initial strategies involving web-based tools like Scansite 2.0 (Obenauer et al., 2003), DISPHOS (Iakoucheva et al., 2004), and KinasePhos 3.0 (Ma et al., 2023) that mainly used scoring methods such as PSSMs. These approaches were sometimes paired with experimental methods like mass spectrometry (Salzano & Crescenzi, 2005; Ramazi & Zahiri, 2021). Despite advances, accurately identifying kinases for specific phosphorylation sites remains a challenge due to poor annotation of the phospho-proteome (Needham et al., 2019). With the expansion of datasets, computational methods utilizing machine learning algorithms such as support vector machines (Jamal et al., 2021), and random forests (Ismail et al., 2016; Liu et al., 2022), significantly improved prediction accuracy. Further advancements came with deep learning, especially convolutional neural networks (CNNs) in models like DeepPhos (Luo et al., 2019) and MusiteDeep (Wang et al., 2017), and more recently, transformer-based models like PhosIDN (Yang et al., 2021) and Phosformer (Zhou et al., 2023) which harnessed transformers to better capture long-range dependencies in protein sequences for enhanced accuracy and interpretability. However, many existing methods overlook the integration of both intra-modality (sequence and structure) and inter-modality (kinase and substrate) information. This limitation is significant, as both types of data are essential for accurate, kinase-specific phosphorylation site prediction.

**Protein Joint Representation Learning.** Recent years have witnessed the emergence of joint representation learning as a promising approach for protein-related tasks. Early attempts like LM-GVP (Wang et al., 2022) sought to combine PLMs with structure encoders. More recent models have leveraged large-scale pre-training on protein sequences while integrating both sequence and structure information (Heinzinger et al., 2023; Zhang et al., 2023; Hu et al., 2023), emerged as powerful tools for various protein-related tasks. Notable among these innovations is SAPROT (Su et al., 2023), which introduced a structure-aware vocabulary for protein language modeling. Similarly, ESM-GearNet (Zhang et al., 2023) has gained significant attention due to its innovative approach, which evaluated and compared various joint representation learning methods for protein sequences and structures. Futhermore, techniques for combining textual and visual information

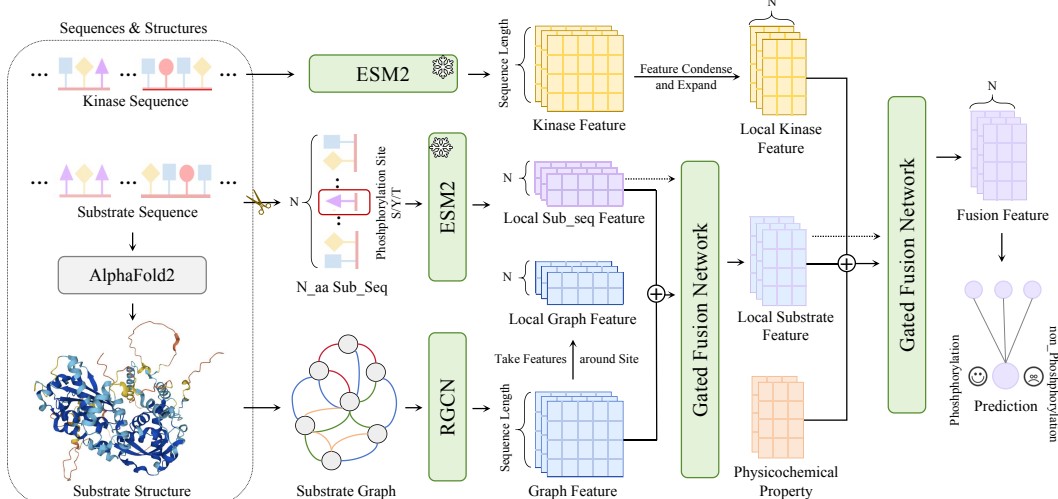

Figure 2: The SAGEPhos model framework.

such as TIRG (Vo et al., 2019) and Vilbert (Lu et al., 2019) have provided valuable insights, inspiring new directions in protein representation learning. Our work builds upon these advancements, developing a joint representation model tailored for phosphorylation tasks.

# 3 METHOD

In addressing the intricate nature of kinase-substrate interactions and the need for effective integration of diverse biological data modalities, we propose a novel approach for phosphorylation site prediction. Our method utilizes inter-modality Bio-Augmented Fusion between kinase and substrate inputs, and further employs intra-modality Bio-Coupled Fusion between sequence and structural information of substrate, as shown in Fig.2. We optimized data preprocessing to enhance embedding quality and introduced a novel fusion framework to dynamically capture effective information from multiple modalities.

## 3.1 PRELIMINARIES

### 3.1.1 NOTATIONS

Let $\mathcal{A} = \{1, \ldots, 20\}$ be the space of amino acid types. A protein sequence of length $N$ is defined as $P = \{s_i\}_{i=1}^{N}$, where $s_i \in \mathcal{A}$. The space of all possible protein sequences is denoted as $\mathcal{P}$.

In phosphorylation prediction, we focus on $n$-mer peptide segments centered around each potentially phosphorylatable residue (serine, threonine, or tyrosine), where $n$ represents the number of amino acids. Let $\mathcal{S}$ be the space of such segments, and $\mathcal{T} \in \mathbb{R}^3$ be the three-dimensional structure space. For each potentially phosphorylatable residue at position $i$, we define:

$$p_i = \{s_{i-(n-1)/2}, \ldots, s_i, \ldots, s_{i+(n-1)/2}\} \in \mathcal{S} \subset \mathcal{A}^n$$
$$T_i = \{t_{i-(n-1)/2}, \ldots, t_i, \ldots, t_{i+(n-1)/2}\} \in \mathcal{T}^n \tag{1}$$

where $p_i$ represents the sequence of the $n$-mer segment, and $T_i$ represents C$\alpha$ atoms of all residues in the $n$-mer segment. The phosphorylation state space $\mathcal{Y} = \{0, 1\}$ indicates phosphorylated (1) or not (0). We denote a kinase protein sequence of length $N_k$ as $K_i = k_j{}_{j=1}^{N_k}$, where $k_j \in \mathcal{A}$. After pooling the kinase features of $K_i$, we broadcast this feature onto the $n$-mer fragment, denoted as $ak_i$. Let $\mathcal{K}$ represent the space of these pooled kinase data, such that $ak_i \in \mathcal{K}$. The joint input space is then defined as $\mathcal{I} = S \times \mathcal{T}^n \times \mathcal{K}$. Consequently, we can formulate the phosphorylation prediction task as learning a function $f : \mathcal{I} \to \mathcal{Y}$ that maps the input space to the phosphorylation state space.

### 3.1.2 PROBLEM FORMULATION

Given a protein sequence $P = \{s_i\}_{i=1}^{N}$ with corresponding 3D structure $T = \{t_i\}_{i=1}^{N}$, where $s_i \in \{1, \ldots, 20\}$ and $t_i \in \mathbb{R}^3$, we aim to predict the phosphorylation state $Y = \{y_i\}_{i=1}^{M}$ for all $M$ potentially phosphorylatable residues, where $y_i \in \{0, 1\}$. For each potentially phosphorylatable residue at position $i$, we consider an $n$-mer peptide segment and its associated features:

$$p'_i = \{(s_j, x_j, e_j, c_j, k)\}_{j=i-(n-1)/2}^{i+(n-1)/2} \tag{2}$$

where $p'_i$ represents the final embedding, $e_j \in \mathbb{R}^d$ denotes a learnable conservation score embedding that is incorporated by adding it to the central phosphorylation site position. $c_j \in \mathbb{R}^d$ represents physicochemical properties, and $k \in \mathbb{R}^d$ indicates the averaged kinase-specific information broadcast to the $n$-mer segment. We aim to learn a function $f : p'_i \to y_i$ for the phosphorylation prediction task by minimizing:

$$\min_{\theta} \mathcal{L}(\theta) = \frac{1}{M} \sum_{i=1}^{M} l(f_\theta(p'_i), y_i) + \lambda R(\theta) \tag{3}$$

where $\theta$ represents the model parameters, $l(\cdot, \cdot)$ is a suitable loss function, and $R(\theta)$ is a regularization term.

### 3.2 INTERACTION-BASED DUAL-MODALITY FATURIZATION

In phosphorylation site prediction, previous models often ignore 3D structural information, but 3D structures include grooves, pockets, and clefts on protein surfaces and interiors, which are helpful for identifying active sites relevant to phosphorylation. Therefore, we introduce structural information to our model. Moreover, since our model focuses on determining whether a specific phosphorylation site can be phosphorylated by a particular kinase, we select functional group around the phosphorylation site as local features relevant to the task. It is worth noting that the prerequisite for these improvements lies in obtaining high-quality embeddings from different modal inputs, making the selection of appropriate encoders paramount.

**Sequence Embedding Acquisition.** Since ESM2 (Lin et al., 2023) has a robust capability in capturing the evolutionary information and nuanced features embedded within protein sequences, we utilize it as encoder for kinase and substrate inputs. It is important to note that kinases exhibit a preference for functional groups surrounding phosphorylation sites, known as peptide specificity, which significantly influence the kinase's recognition and catalytic efficiency (Fujii et al., 2004; Miller & Turk, 2018), thus we extract the local sequences around specific phosphorylation sites in substrates. The embeddings for substrate and kinase sequences are obtained as follows:

$$X_{sSeq} = f_\theta(g_\sigma(S_{sub})) \qquad\qquad X_{kSeq} = f_\theta(h_{\theta_k}(g_\sigma(S_{kin}))) \tag{4}$$

Here, $S_{sub}$ and $S_{kin}$ represent the local $n$-mer substrate and overall kinase sequences, respectively, where $n$ is the number of amino acids. $g_\sigma(\cdot)$ is the frozen ESM2 encoder applied to both sequences. The function $h_{\theta_k}(\cdot)$ processes kinase sequences by averaging features and broadcasting them, while $f_\theta(\cdot)$ projects encoded sequences into a relevant feature space.

**Structure Embedding Acquisition.** As AlphaFold2 (Jumper et al., 2021) can highly predict accurate protein structure, we use it to capture the substrate structures for phosphorylation site prediction. We further encode the structures utilizing Relational Graph Convolutional Network (R-GCN) (Schlichtkrull et al., 2018), which is designed to handle various relationships and can effectively model the intricate dependencies within the protein structure. For details of the structural encoder selection process, please refer to Appendix F. The embedding for each node $v$ in the R-GCN is computed as:

$$h_v(l+1) = \sigma \left( W_0(l) h_v(l) + \sum_{r \in \mathcal{R}} \sum_{u \in \mathcal{N}_v^r} \frac{1}{|\mathcal{N}_v^r|} W_r(l) h_u(l) + b(l) \right) \tag{5}$$

where $h_v^{(l)}$ is the embedding of node $v$ at layer $l$, $\mathcal{R}$ is the set of relations, $\mathcal{N}_v^r$ denotes the set of neighbor nodes of node $v$ under relation $r$, $W_0^{(l)}$, $W_r^{(l)}$ are learnable weight matrices, $b^{(l)}$ is a bias term, and $\sigma$ is RELU, an activation function.

In the realm of substrate structure, the employment of short peptides serves to prevent the masking or interference of phosphorylation sites by the intricate tertiary structure of the complete protein, as evidenced in prior studies (Johnson & Lewis, 2001), complementing the concept of peptide specificity. Consequently, we propose a novel function for local structural feature extraction to achieve a structure representation around phosphorylation sites aligning with the substrate sequence in the following manner:

$$X_{sStru} = \text{READOUT}(h_v^{(L)} | v \in \mathcal{V}_{i-(n-1)/2:i+(n-1)/2}) \tag{6}$$

where $X_{stru}$ is the local structural feature, $L$ is the index of the final R-GCN layer, $i$ is the index of the phosphorylation site and $\mathcal{V}_{i-(n-1)/2:i+(n-1)/2}$ represents the set of nodes corresponding to the $n$ amino acids centered on the phosphorylation site, aligning with the local sequence information. The Readout function can be defined as:

$$\text{READOUT}(h_v^{(L)}) = [\text{Mean}(h_v^{(L)} | v \in \mathcal{V}_j)], j = i - (n-1)/2, \ldots, i + (n-1)/2 \tag{7}$$

where Mean is the average pooling operation over the nodes corresponding to each amino acid. This READOUT function first averages the node embeddings for each amino acid and then concatenates these averaged embeddings for the $n$ amino acids in the local window, ensuring that the proposed method captures local substrate structures that aligns with local sequences.

### 3.3 MULTIMODAL GATED-RESIDUAL INTEGRATION MODULE

Traditional computational models often focus on kinase families, overlooking the interactions between individual kinases and substrates. Meanwhile, current mainstream approaches, while considering individual kinases, solely emphasize sequence modalities, neglecting crucial structural information. In response to these constraints, we present the Multimodal gatEd-Residual inteGration modulE (MERGE), which integrates substrate sequence, substrate structure, and kinase sequence modalities, as shown in Fig.3. By fusing kinase inputs, MERGE provides more precise targets for kinase inhibitors, and through the incorporation of substrate structural data, it enhances understanding of the local environment surrounding phosphorylation sites, such as surface accessibility. The core of MERGE is a Dynamic fusiOn meChanism (DOC), which defined fusion feature as follows:

$$X_{\text{f}} = \alpha \cdot \boldsymbol{\Phi}(X_p, X_a) + \beta \cdot \boldsymbol{\Psi}(X_p, X_a) \tag{8}$$

DOC integrates both a gated mechanism $\boldsymbol{\Phi}$ and a residual mechanism $\boldsymbol{\Psi}$, operating on two distinct input modalities: the primary modality $X_p$ and the auxiliary modality $X_a$. The learnable parameters $\alpha$ and $\beta$ fine-tune the contributions of these mechanisms, allowing for adaptive fusion of the multimodal inputs. The gating mechanism $\boldsymbol{\Phi}$ is formulated as:

$$\boldsymbol{\Phi}(X_p, X_a) = \sigma\left(\mathbf{W}_{g2} \cdot \text{ReLU}(\mathbf{W}_{g1} * (\begin{bmatrix} X_p \\ X_a \end{bmatrix} + E_{site}))\right) \odot X_p \tag{9}$$

where $E_{site}$ is an embedding vector that highlights the phosphorylation site by an element-wise addition to the encoded substrate at the phosphorylation site. And $\sigma$ denotes the sigmoid activation function, $\odot$ represents the element-wise multiplication, $*$ indicates a linear transformation followed by batch normalization, and $\mathbf{W}_{g1}$, $\mathbf{W}_{g2}$ are trainable weight matrices. The gated mechanism $\boldsymbol{\Phi}$ filter out less task-relevant information from the auxiliary modality, while simultaneously retaining the primary modality feature. Moreover, the residual mechanism $\boldsymbol{\Psi}$ is defined as:

$$\boldsymbol{\Psi}(X_p, X_a) = \mathbf{W}_{r2} \cdot \text{ReLU}(\mathbf{W}_{r1} * (\begin{bmatrix} X_p \\ X_a \end{bmatrix} + E_{site})) \tag{10}$$

where $\mathbf{W}_{r1}$ and $\mathbf{W}_{r2}$ are also learnable weight matrices. In contrast to the gated network, the residual mechanism $\boldsymbol{\Psi}$ harnesses information from the auxiliary modality to refine the primary modality space, thereby enriching the representation of the primary modality.

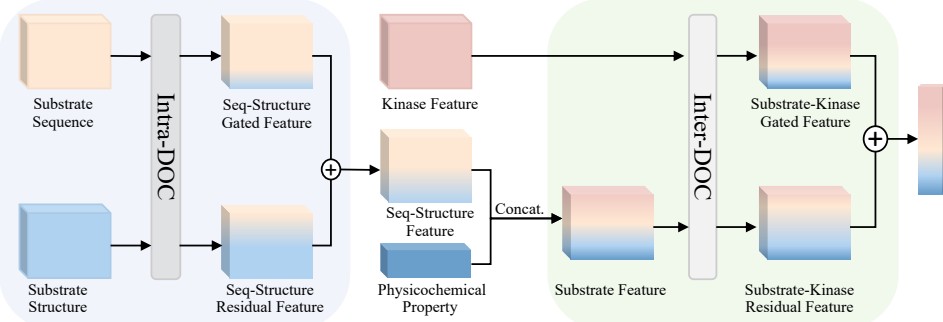

Figure 3: A two-step multi-modal feature fusion process. DOC is a dynamic fusion mechanism.

MERGE integrates multiple modalities through a two-stage fusion process, employing the proposed mechanism, DOC, in each stage. In the first stage, we fuse the substrate sequence and structure information:

$$X_{sub} = \alpha_1 \cdot \boldsymbol{\Phi}_1(X_{sSeq}, X_{sStru}) + \beta_1 \cdot \boldsymbol{\Psi}_1(X_{sSeq}, X_{sStru}) \tag{11}$$

where $X_{sSeq}$ represents the local substrate sequence features surrounding the phosphorylation site, $X_{sStru}$ denotes the local substrate structural features, and $X_{sub}$ is the fused local substrate representation. Given that local sequence information is inherently accurate, while structural information is predicted and potentially less reliable, we adopt the sequence input as the primary modality and utilize the structural input to refine and enrich the sequence feature space. This Bio-Augmented Fusion method allows us to inject valuable spatial accessibility information into the local sequence context while filtering out potentially inaccurate structural noise. To further capture the interplay between substrates and kinases, we combine the fused substrate features with kinase sequence features and physicochemical properties in the second stage of MERGE:

$$X_{fus} = \alpha_2 \cdot \boldsymbol{\Phi}_2\left(X_{sub}, \begin{bmatrix} X_{kSeq} \\ X_{phy} \end{bmatrix}\right) + \beta_2 \cdot \boldsymbol{\Psi}_2\left(X_{sub}, \begin{bmatrix} X_{kSeq} \\ X_{phy} \end{bmatrix}\right) \tag{12}$$

where $X_{kSeq}$, $X_{phy}$, and $X_{fus}$ represent kinase sequence features, physicochemical properties, and the final fused representation, respectively. We designate the substrate as the primary modality for three reasons: 1) Substrate features are numerous and concentrated around the phosphorylation site, demonstrating stronger task relevance. 2) In contrast, kinases are fewer in number and are represented by full-length sequences without specific reaction sites, which can contain some some task-irrelevant noise. 3) Physicochemical properties complement substrate amino acids without overshadowing them. Thus kinases and physicochemical properties serve as auxiliary modalities. By using the smaller kinase space to modify the larger substrate space, we aim to identify their common space. This Bio-Coupled Fusion approach enables us to establish a multi-scale representation that captures the kinase-substrate interaction information.

These mechanisms allow MERGE to adaptively integrate task-relevant information from multiple modalities, capturing kinase-substrate interactions and improving the overall predictive performance of the model.

## 4 EXPERIMENT

**Datasets.** Our phosphorylation site prediction dataset was compiled from Phospho.ELM (Dinkel et al., 2010), PhosphoNetworks (Hu et al., 2014), and PhosphoSitePlus (Hornbeck et al., 2012), with structural information incorporated from AlphafoldDB (Jumper et al.). After rigorous curation and filtering, the final dataset comprises 18,360 positive samples. Additionally, we modified the split methods to simulate a cold-start scenario (Zhu et al., 2021), introducing entirely new kinase or substrate sequences in the test set, denoted respectively as Kinase-cold-start and Substrate-cold-start. [1] For database sources, dataset construction, and partitioning methods, see Appendix A.

---

[1]Due to the high interconnectivity (99.9%) in our dataset, with most sites linked to multiple kinases and vice versa, creating a truly cold-start test set with completely unseen kinases and substrates was not feasible.

**Evaluation Metrics and Hyperparameter Settings.** We evaluated model performance using Accuracy, Area Under the Receiver Operating Characteristic Curve (AUC-ROC) and Area Under the Precision-Recall Curve (AUC-PRC). Additionally, we included the False Positive Rate (FPR) to specifically assess the model's tendency to incorrectly identify non-phosphorylation sites. Due to space limitations, implementation details and hyperparameters for each dataset and partition are provided in Appendix B.

**Baselines.** Recent phosphorylation site predictors vary in input requirements and prediction targets. MusiteDeep (Wang et al., 2017) predicts phosphorylation potential for all sites in a full-length substrate sequence. DeepPhos (Luo et al., 2019), PhosIDN (Yang et al., 2021), and PhosIDNSeq focus on local sequence context without considering individual kinase information. EMBER (Kirchoff & Gomez, 2022) predicts which kinase families can phosphorylate a given substrate. Most relevant to our work, Phosformer (Zhou et al., 2023) and Phosformer-ST (Zhou et al., 2024) predict whether a local sequence can be phosphorylate by a individual kinase.

## 4.1 PERFORMANCE COMPARISON

Table 1: Comparison of our model SAGEPhos with existing phosphorylation predictors on warm-start dataset. Results in **bold** and underlined are the top-1 and top-2 performances, respectively.[3]

| Method | Warm-start Partition | | | |
| --- | --- | --- | --- | --- |
| | Acc↑ | AUC-ROC↑ | AUC-PRC↑ | FPR↓ |
| MusiteDeep | 73.5±0.3 | 78.5±0.1 | 7.3±0.1 | 26.4±0.3 |
| DeepPhos | 58.5±0.4 | 61.9±0.7 | 62.0±0.2 | 44.2±3.0 |
| PhosIDNSeq | 51.8±0.0 | 50.4±0.2 | 51.8±1.8 | 89.1±0.0 |
| PhosIDN | 51.6±0.1 | 50.9±0.1 | 51.2±0.1 | 93.7±0.5 |
| EMBER | 53.4±2.1 | 52.5±0.4 | 5.9±0.1 | 46.4±2.3 |
| Phosformer | 54.6 | 72.7 | 73.6 | **1.4** |
| Phosformer-ST | 64.7 | 78.9 | 79.0 | 95.6 |
| SAGEPhos (Ours) | **80.6**±0.2 | **88.3**±0.1 | **86.2**±0.2 | 21.7±0.4 |

**Baselines Comparison.** SAGEPhos surpasses existing phosphorylation site predictors across various evaluation metrics in warm start dataset, achieving a notable 10% and 12% improvements in prediction accuracy and AUC-ROC, as detailed in Table.1. Its success lies in merging structural insights and kinase-substrate interactions through advanced two-stage selective integration, enriching phosphorylation site predictions well beyond traditional models. While SAGEPhos does not achieve the lowest False Positive Rate (FPR) at 23.5%, it maintains balanced excellence in accuracy and AUC scores, contrasting Phosformer's low FPR (1.4%) and TPR (11.3%), which suggests a tendency to predict negatives. To further assess the model's performance, we conducted comparisons with other baseline models on cold start datasets, as illustrated in Table.2. Our model surpasses the performance of other models in both kinase and substrate cold starts. Interestingly, in comparison to warm start data, substrate cold start dataset hardly affected performance, indicating our model's capability to infer phosphorylation site knowledge from lengthy substrate sequences, enhanced by structural data, even if without prior exposure. In contrast, kinase cold starts showed a performance drop, attributed to the sparse kinase data, complex sequences, and the absence of entire kinase families from the training set, which could impede the model's understanding and performance.

**Robustness.** In assessing our model's robustness, we examined its performance across various dataset partition schemes (80:10:10, 75:15:15, 60:20:20, 50:25:25, and 40:30:30 for train:validation:test). As illustrated in Fig.4(a), performance slightly declines at 50:25:25 and 40:30:30 splits. Notably, even under these data-constrained conditions, our model consistently outperforms benchmark models trained on the standard 80:10:10 split. This demonstrates our model's robustness and its ability to maintain reliable results with limited training data.

---

[3]EMBER only allows family-level predictions. And the training code for Phosformer and Phosformer-ST is not open-sourced, so their performance may be inflated as we cannot verify potential overlap between their training data and our test set when evaluating them.

Table 2: Comparison of our model SAGEPhos with existing kinase-specific phosphorylation predictors on cold-start datasets.

| Method | Kinase-cold-start Partition | | | | Substrate-cold-start partition | | | |
|---|---|---|---|---|---|---|---|---|
| | Acc↑ | AUC-ROC↑ | AUC-PRC↑ | FPR↓ | Acc↑ | AUC-ROC↑ | AUC-PRC↑ | FPR↓ |
| MusiteDeep | 68.3±3.5 | 73.8±0.1 | 6.4±0.1 | 31.7±3.6 | 67.5±1.7 | 22.2±0.4 | 0.8±0.1 | 31.8±1.7 |
| DeepPhos | 55.7±0.5 | 57.8±0.6 | 58.5±0.6 | 51.9±5.3 | 59.5±0.4 | 64.0±0.3 | 60.6±0.3 | 33.9±1.9 |
| PhosIDNSeq | 55.0±0.0 | 48.5±2.0 | 49.4±1.8 | 89.1±0.0 | 52.5±0.0 | 52.2±1.5 | 52.8±1.3 | 89.1±0.0 |
| PhosIDN | 55.0±0.1 | 51.1±0.1 | 50.7±0.2 | 89.1±0.1 | 52.3±0.1 | 51.0±0.1 | 51.4±0.1 | 92.2±0.4 |
| EMBER | 52.0±3.0 | 48.6±2.2 | 4.8±0.4 | 4.8±3.6 | 49.4±4.1 | 49.5±1.8 | 6.2±0.2 | 50.6±5.0 |
| Phosformer | 55.7 | 72.3 | 73.2 | **1.5** | 55.2 | 74.9 | 75.7 | **1.6** |
| Phosformer-ST | 61.0 | 72.7 | 73.3 | 94.9 | 63.5 | 79.3 | 79.7 | 97.0 |
| SAGEPhos (Ours) | **68.6**±0.8 | **76.9**±0.3 | **75.9**±0.3 | 22.4±0.6 | **79.1**±0.4 | **87.2**±0.2 | **85.3**±0.4 | 17.1±0.4 |

## 4.2 KINASE-SPECIFIC STUDY

**Zero-shot Prediction.** A key strength of our model is its ability to generalize to new distributions. To demonstrate this capability, we conducted an experiment using CDK17, a kinase absent from our training set, along with its corresponding substrates sourced from an independent dataset (Johnson et al., 2023). As illustrated in Fig.4(b), our model significantly outperforms baselines in predicting phosphorylation sites for the unseen kinase-substrate pair, highlighting its robust adaptability to novel scenarios. Similar experiments on four additional unseen kinases further validated our model's generalizability. Detailed experimental procedures are provided in the Appendix C.

**Embedding Visualization.** We used t-SNE (Van der Maaten & Hinton, 2008) to visualize our model's learned embedding. The final epoch (Fig.4(e)) shows clearer kinase family-based clustering compared to the first epoch (Fig.4(d)). This reveals our model's capacity to capture evolutionary relationships among kinase groups, highlighting its effectiveness in integrating key features, filtering noise, and distilling essential biochemical and evolutionary signals defining kinase function.

## 4.3 CASE STUDY

To understand our model's capacity to capture substrate specificity and contextual information in phosphorylation site prediction, we conducted a case study on two kinases: GSK3B and MK01. We selected 10 substrate peptides for each kinase and visualized the importance of amino acids around the phosphorylation sites.

As shown in Fig.4(c), our model emphasizes phosphorylation sites, shown by deep red color blocks. Additionally, intense coloration of serine, threonine, or tyrosine (S/T/Y) at non-phosphorylation sites indicates learned spatial information and potential alternative sites. For GSK3B specifically, deeply colored S/T/Y residues near phosphorylation sites reflect the model's recognition of the "priming" phosphorylation pattern and priming phosphorylation. Meanwhile, in the case of MK01, highlighted proline residues suggest the model's awareness of the "proline-directed phosphorylation" pattern, crucial for proline-directed phosphorylation. Fig.4(c) displays a subset of the analyzed sequences, and full dataset and detailed analysis are in Appendix D.

## 4.4 ABLATION STUDY AND HYPERPARAMETER SENSITIVITY

**Ablation Study.** To investigate the impact of our residual and gated fusion model and phosphorylation site emphasis on performance, we compared vanilla SAGEPhos with four variants: *(A)* w/o fusion: neither intra- (substrate sequence and substrate structure) fusion model nor inter- (substrate and kinase)

Table 3: Ablation study on various modules.

| Method | Accuracy↑ | AUC-ROC↑ | AUC-PRC↑ | FPR ↓ |
|---|---|---|---|---|
| w/o fusion | 65.8±0.7 | 73.1±0.8 | 73.0±0.4 | 41.0±0.8 |
| w/- intra fusion | 78.2±0.5 | 85.8±0.3 | 83.5±0.3 | 27.4±0.1 |
| w/- inter fusion | 77.6±0.2 | 86.0±0.2 | 83.8±0.4 | 27.9±1.4 |
| w/- fusion & w/o empha | 76.8±0.2 | 84.5±0.3 | 82.6±0.3 | 27.2±0.2 |
| w/- fusion & w/- empha | **80.6**±0.2 | **88.3**± 0.1 | **86.2**±0.2 | **21.7**±0.4 |

fusion model, and simple concatenation is used instead of fusion; *(B)* w/- intra fusion: only intra-

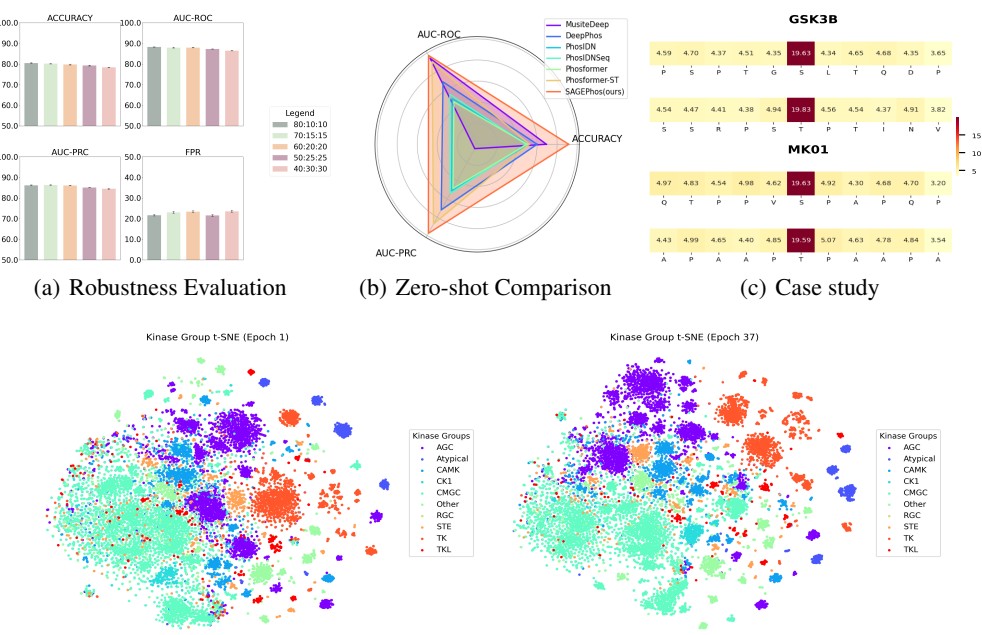

(a) Robustness Evaluation    (b) Zero-shot Comparison    (c) Case study

(d) Kinase-group Visualization at First Epoch    (e) Kinase-group Visualization at Final Epoch

Figure 4: (a) Assessing model robustness using various dataset partition ratios. (b) Evaluating the model's ability to predict interactions between CDK17 (a kinase unseen in the training set) and its corresponding substrates, compared to other benchmark models. (c) Visualizing the importance of amino acids surrounding each phosphorylation site. (d) t-SNE projection after the first epoch, re-colored by kinase groups. (e) t-SNE projection after the final epoch, re-colored by kinase groups.

fusion model; *(C)* w/- inter fusion: only inter-fusion model; *(D)* w/- fusion and w/o empha: both inter-fusion model and intra-fusion model, but no emphasis on phosphorylation sites. The results in Table 3 reveal: (1) Comparing (B) to (A), the intra-fusion model improves performance, indicating that structural information complements sequence data more effectively than simple concatenation. This fusion mechanism filters out inaccurate structural information while enhancing the sequence space representation. (2) Comparing (C) to (A), the inter-fusion model enhances performance. This suggests that our fusion mechanism, by modifying the substrate space with kinase information, captures more nuanced interaction details than simple concatenation. (3) Emphasizing phosphorylation sites boosts performance, especially in AUC-PRC, by enabling the model to focus on critical areas and better understand the phosphorylation process.

**Hyperparameter Sensitivity.** We explored how the initial values of the gated feature and residual feature weights—$\alpha_1$, $\alpha_2$, $\beta_1$, and $\beta_2$—influence the sensitivity of two fusion modules. As detailed in Appendix E, Our analysis shows the model is not highly sensitive to initial hyperparameter values, suggesting architectural robustness. This indicates flexibility and reliability across varying initial conditions, with performance not heavily dependent on precise starting weights.

## 5  CONCLUSION

In this study, we introduce a novel dataset combining substrate sequence and structure with kinase sequence information. Moreover, we propose a multi-modal gated and residual integration module, featuring bio-coupled and bio-augmented fusion stages, to dynamically learn critical information from each modality. Experimental results demonstrate superior performance over benchmarks across multiple datasets. Moreover, our model exhibits strong robustness and generalizability, performing well on test sets with diverse distributions.

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

## A   DATASETS AND PARTITION METHOD

Our phosphorylation site prediction dataset was compiled from multiple comprehensive resources:

(1) Phospho.ELM (Dinkel et al., 2010): A curated database of experimentally verified phosphorylation sites in eukaryotic proteins. Phospho.ELM version 2.0 contains 1,703 phosphorylation site instances for 556 phosphorylated proteins.

(2) PhosphoNetworks (Hu et al., 2014): A high-resolution phosphorylation network database integrating protein microarray-verified kinase-substrate relationships (KSRs) and MS-verified phosphorylation sites. It includes 24,046 raw KSRs and 3,656 refined KSRs, along with 300 novel predicted phosphorylation motifs.

(3) PhosphoSitePlus (Hornbeck et al., 2012): An extensive, manually curated resource of post-translational modifications. It comprises 129,082 non-redundant sites on 14,256 non-redundant proteins, with over 90% of these sites from human and mouse.

(4) AlphaFoldDB (Jumper et al.): A database provides predicted 3D structures for proteins. The latest release contains over 200 million entries, covering a broad range of UniProt proteins. We

incorporated this structural information to enhance our dataset with spatial context for the phosphorylation sites.

In the process of consolidating the first three databases, we meticulously verified each data point by cross-referencing the UniProt Database, focusing specifically on human phosphorylation sites and their associated kinases. Subsequently, we searched for corresponding substrate structures in AlphaFoldDB, retaining entries with available structures and excluding those without. Following a rigorous curation process to eliminate redundancies and ensure data quality, coupled with careful filtering to maintain consistency across sources, our final dataset comprises 18,360 positive samples. For each kinase, we selected negative samples from two sources: (1) known phosphorylation sites that are validated substrates of other kinases, and (2) experimentally verified phosphorylatable sites lacking reported kinase associations, resulting in a balanced 1:1 ratio of positive to negative samples.

To facilitate robust model training and evaluation, we partitioned the dataset into training, validation, and test sets using a ratio of 8:1:1. This approach ensures comprehensive assessment of our model's performance and generalization.

Additionally, we modified the split methods to simulate a cold-start scenario (Zhu et al., 2021), introducing entirely new kinase or substrate sequences in the test set, denoted respectively as Kinase-cold-start and Substrate-cold-start. Specifically, for kinase cold-start, dataset splitting was performed based on kinase identity. All data points sharing the same kinase were assigned to the same set to ensure no kinase overlap between training, validation, and testing sets; and for substrate cold-start, dataset splitting was based on substrate 11-mer sequence identity. All data points with identical substrate sequences were assigned to the same set to avoid sequence overlap between training, validation, and testing sets.

## B  IMPLEMENTATION DETAILS AND HYPERPARAMETERS

The substrate local sequence is represented by an 11-mer and encoded using the ESM-2-650M protein language model to obtain sequence embeddings. The Substrate Structure Encoder, implemented as a Relational Graph Convolutional Network (RGCN), utilizes 6 layers with a hidden dimension of 512. We employ four fundamental physicochemical properties commonly used in protein structure-function analysis: (1) Aliphatic, (2) Aromatic, (3) Acidic charged, and (4) Basic charged. Each property is represented as a binary category. We set the learning rate to 1e-5 and weight decay to 1e-4, with training conducted over 100 epochs. The predictor, a Multi-Layer Perceptron (MLP), consists of 3 layers with a dropout rate of 0.2. The learnable weights for both the gated module and residual module $\alpha_1$, $\alpha_2$, $\beta_1$, and $\beta_2$ range from 0.1 to 1.0. Our experiments were implemented using PyTorch 2.2.2, leveraging an Intel(R) Xeon(R) Gold 6426Y CPU and 2 NVIDIA A40 GPUs for computational resources.

## C  ZERO-SHOT PREDICTION DETAILS

Recently, a new phosphorylation site prediction dataset has emerged (Johnson et al., 2023), ranking 303 kinases based on their likelihood to catalyze phosphorylation for a given substrate with 15 amino acids surrounding phosphorylation site. Kinases ranked at the top are considered capable of catalyzing phosphorylation, while those at the bottom are deemed unlikely to do so. To further validate our model's generalizability, we extracted data for the CDK17 kinase from the new dataset, which was not present in our original training set. We used this data to create a new test set, where substrate-kinase pairs ranked first for CDK17 were treated as positive samples, and those ranked the bottom two were designated as negative samples. We then evaluated our model, trained exclusively on our original dataset, on this new CDK17-specific test set. The re-

Table A1: Comparison of our model SAGEPhos with existing phosphorylation predictors on CDK17 dataset. Results in **bold** and underlined are the top-1 and top-2 performances, respectively.

| Method | Acc↑ | AUC-ROC↑ | AUC-PRC↑ |
|---|---|---|---|
| MusiteDeep | 69.3 | 94.2 | 4.8 |
| DeepPhos | 60.0 | 68.4 | 71.9 |
| PhosIDNSeq | 48.9 | 54.7 | 53.8 |
| PhosIDN | 51.0 | 50.0 | 51.0 |
| Phosformer | 50.0 | 46.5 | 48.4 |
| Phosformer-ST | 52.2 | 90.4 | 87.4 |
| SAGEPhos (Ours) | **91.6** | **97.4** | **97.6** |

sults, as shown in Table.A1, demonstrate that our model significantly outperforms other benchmark models in predicting CDK17-mediated phosphorylation sites.

And we then expanded our evaluation to include four additional kinases that were not present in our original training set (MYLK4, CDKL1, PHKG2, SRPK3). The four kinases were extracted from the same new phosphorylation site prediction dataset as CDK17. And we compared SAGEPhos's performance with MusiteDeep, which is the most robust and best-performing baseline among those we used, shown in Table.A2. The result shows that SAGEPhos demonstrates stable and excellent performance across all 5 kinase datasets, ACC metrics consistently remain between 82%-0.88%, AUROC and AUPRC metrics mostly achieve high scores above 90%. Notably, while MusiteDeep achieves comparable AUC-ROC scores, its significantly lower AUC-PRC values (4.0-23.8%) indicate poor performance in identifying true phosphorylation sites in highly imbalanced real-world scenarios. In contrast, SAGEPhos demonstrates consistent superior performance across all metrics, showing robust generalization ability in precisely detecting phosphorylation sites.

This demonstrate that SAGEPhos maintains good predictive performance even on unseen kinase families and generalizes well.

Table A2: Comparison of our model SAGEPhos with MusiteDeep on zero-shot datasets.

| Method | SAGEPhos | | | MusiteDeep | | |
|---|---|---|---|---|---|---|
| | Acc↑ | AUC-ROC↑ | AUC-PRC↑ | Acc↑ | AUC-ROC↑ | AUC-PRC↑ |
| MYLK4 | 84.4 | 97.9 | 98.1 | 71.1 | 94.5 | 23.8 |
| CDKL1 | 82.6 | 90.0 | 91.0 | 71.1 | 91.1 | 4.0 |
| PHKG2 | 88.2 | 95.8 | 95.3 | 71.8 | 90.3 | 15.2 |
| SRPK3 | 81.3 | 88.9 | 88.2 | 70.6 | 95.6 | 8.7 |

## D  CASE STUDY DETAILS

To assess our model's ability to capture substrate specificity and contextual information in phosphorylation site prediction, we conducted a case study focusing on two kinases: GSK3B and MK01. GSK3B is a multifunctional serine/threonine protein kinase whose dysregulation is associated with various disorders, including neurodegenerative diseases (such as Alzheimer's) and bipolar disorder. Typically active, GSK3B's activity is regulated through inhibitory phosphorylation. It shows a preference for phosphorylating substrates that have been previously phosphorylated by other kinases. MK01, a crucial member of the Mitogen-Activated Protein Kinase (MAPK) family, plays a central role in cell signal transduction. It requires dual phosphorylation by upstream kinases (like MEK) for activation and preferentially phosphorylates substrates containing the specific sequence motif (P-X-S/T-P).

We selected 10 substrate peptides for each kinase and visualized the importance of amino acids surrounding the phosphorylation sites. As illustrated in Fig.A1, our analysis revealed several key findings: (1) Our model primarily emphasizes critical phosphorylation site information, as evidenced by the deep red color blocks at phosphorylation sites. (2) Notably, some S/T/Y residues at non-phosphorylation sites also display intense coloration, suggesting that the model has learned to incorporate spatial information and can identify other potential phosphorylation sites within the sequences. (3) For GSK3B, we observed deeply colored phosphorylatable residues either preceding or following the phosphorylation site in most of its catalyzed sequences. This indicates that our model has detected additional potential phosphorylation sites, aligning with GSK3B's known preference for phosphorylating primed substrates (referred to as "priming" phosphorylation). GSK3B typically recognizes the (S/T)XXX(S/T) sequence pattern, where the second S/T is the phosphorylation site, and the first S/T is often pre-phosphorylated. (4) In the case of MK01, we frequently observed deeply colored proline (P) residues. This suggests that our model has, to some extent, recognized the "proline-directed phosphorylation" pattern (P-X-S/T-P), which is crucial for accurately predicting MK01-catalyzed substrate phosphorylation sites.

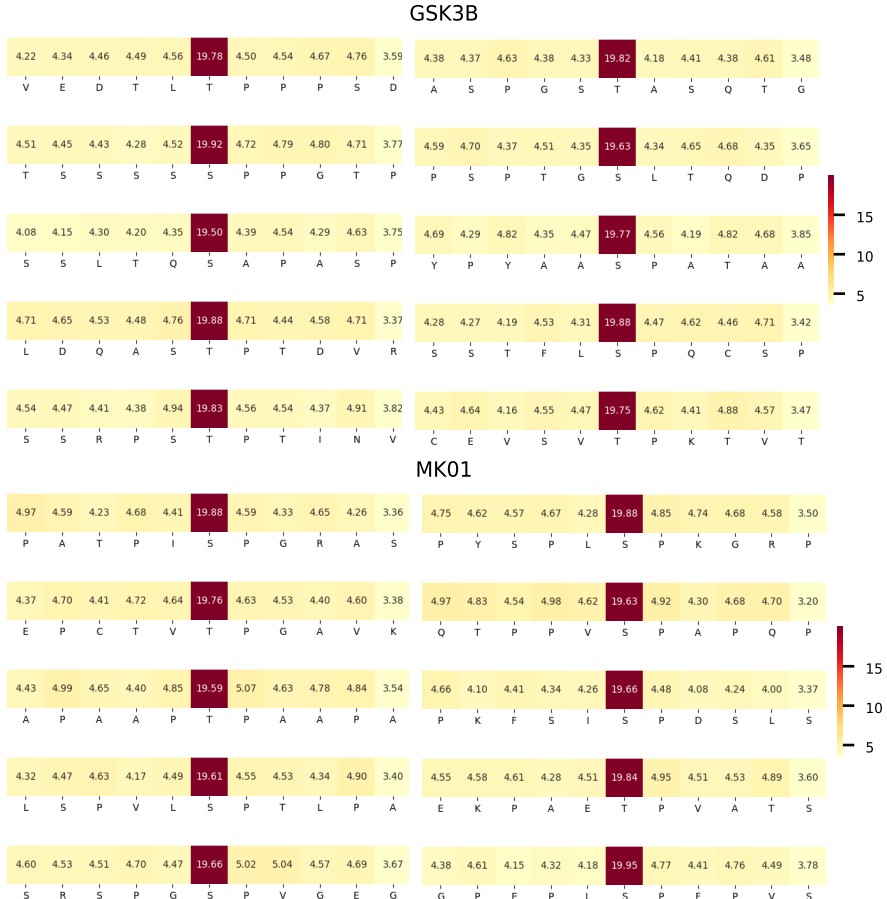

Figure A1: The importance of amino acids around the phosphorylation sites on GSK3B and MK01.

These findings demonstrate our model's sophisticated ability to capture nuanced patterns in kinase-substrate interactions, reflecting both the general mechanisms of phosphorylation and the specific preferences of individual kinases.

Table A3: Performance with various initial weights.

| Weights | | | | Accuracy↑ | AUC-ROC↑ | AUC-PRC↑ | FPR↓ |
|---|---|---|---|---|---|---|---|
| $\alpha_1$ | $\beta_1$ | $\alpha_2$ | $\beta_2$ | | | | |
| 0.1 | 1.0 | 1.0 | 1.0 | 79.1 | 86.6 | 84.3 | 26.0 |
| 0.5 | 1.0 | 1.0 | 1.0 | 80.1 | 87.1 | 84.6 | 24.2 |
| 1.0 | 1.0 | 1.0 | 1.0 | 79.5 | 87.1 | 85.0 | **23.5** |
| 1.0 | 0.5 | 1.0 | 1.0 | 80.0 | 87.4 | 85.3 | 24.2 |
| 1.0 | 0.1 | 1.0 | 1.0 | 79.5 | 87.6 | 85.3 | 26.2 |
| 1.0 | 0.5 | 0.1 | 1.0 | **80.2** | **87.8** | 85.2 | 25.7 |
| 1.0 | 0.5 | 0.5 | 1.0 | **80.2** | 87.6 | **85.6** | 24.8 |
| 1.0 | 0.5 | 1.0 | 0.5 | 79.6 | 87.2 | 85.4 | 25.3 |
| 1.0 | 0.5 | 1.0 | 0.1 | 77.9 | 85.9 | 85.2 | 25.7 |

## E  HYPERPARAMETER SENSITIVITY

We explored how the initial values of the gated feature and residual feature weights—$\alpha_1$, $\alpha_2$, $\beta_1$, and $\beta_2$—influence the sensitivity of two fusion modules. As shown in Table.A3, our model is not

highly sensitive to initial hyperparameter values, suggesting architectural robustness. This indicates flexibility and reliability across varying initial conditions, with performance not heavily dependent on precise starting weights.

## F  STRUCTURE ENCODER SELECTION DETAILS

We conducted a comprehensive comparison of various graph neural networks on our structural graphs, including Graph Convolutional Network (GCN), Graph Isomorphism Network (GIN), Graph Attention Network (GAT), and Relational Graph Convolutional Network (R-GCN).

As illustrated in Table A4, R-GCN achieves superior performance compared to simpler baseline models (GCN, GIN, and GAT). R-GCN consistently outperforms these alternatives across all metrics. GCN's uniform message passing, GIN's structure learning, and GAT's attention mechanism alone are insufficient for capturing the complex residue relationships, as evidenced by their lower performance. R-GCN uses relation-specific transformation matrices to effectively capture and learn the distinct importance of different types of residue relationships (sequential, spatial, and k-nearest neighbor connections) in phosphorylation site prediction.

Table A4: Comparison of various graph neural networks.

| Method | Acc↑ | AUC-ROC↑ | AUC-PRC↑ | FPR↓ |
|--------|------|----------|----------|------|
| GCN | 78.2 | 86.1 | 84.3 | 25.5 |
| GIN | 78.6 | 85.9 | 84.2 | 23.8 |
| GAT | 77.6 | 86.0 | 84.2 | 27.5 |
| R-GCN | **80.6** | **88.3** | **86.2** | **21.7** |

