# OpenReview forum: "SAGEPhos: Sage Bio-Coupled and Augmented Fusion for Phosphorylation Site Detection"
_ICLR.cc/2025/Conference — ICLR 2025 Poster_

### Official Review · Reviewer_1xkZ · 2024-10-30

**Soundness:** 3
**Presentation:** 3
**Contribution:** 3
**Rating:** 6
**Confidence:** 3

**Summary:**

The paper introduces SAGEPhos, a novel structure-aware framework designed to enhance phosphorylation site prediction by integrating sequence and structural information in a multi-modal fusion approach. To address the limitations of current kinase-specific prediction methods that primarily focus on sequence data, SAGEPhos employs two complementary fusion strategies: Bio-Coupled Fusion, which refines substrate feature spaces using kinase sequence data for inter-modality integration, and Bio-Augmented Fusion. And these methods also provide a shared semantic space that captures crucial kinase-substrate interaction patterns. Besides, the authors contribute a new phosphorylation site prediction dataset enriched with structural information, establishing a more comprehensive benchmark for this domain. Experimental results show that SAGEPhos achieves significant improvements over existing approaches in both warm-start and zero-shot scenarios.

**Strengths:**

1) The paper is well-written.

2) The proposed method achieves state-of-the-art performance on new phosphorylation site prediction dataset  compared to baselines.

3) The paper introduces a novel method to phosphorylation site prediction by integrating both sequence and structural data through Bio-Coupled and Bio-Augmented Fusion modules. This dual fusion approach is particularly innovative, allowing SAGEPhos to capture additional information of kinase-substrate interactions, addressing an limitation in existing models that focus on sequence data.

4) The new dataset with added structural information is a valuable addition, offering a benchmark supports future work in this field.

**Weaknesses:**

1) The paper uses structural information as an auxiliary modality fused with sequence data; however, it lacks an in-depth discussion on the mechanisms of multi-modal interaction. Specifically, after R-GCN extracts structural features, the paper does not clarify how these features are effectively leveraged sequence information in specific task, such as complex kinase-substrate interactions.

2) The addition of new structural data as a modality increases the complexity of the SAGEPhos model. However, the paper omits discussion of the additional computational costs introduced by this additional complexity.

3) Though the new dataset with structural information is a valuable contribution, it remains relatively small and limited in diversity, especially in terms of kinase families and phosphorylation patterns.

**Questions:**

1) Do you consider directly comparing SAGEPhos with a model that simply combines sequence and structure data? It seems  the improvements in SAGEPhos may result more from the utilization of structural information itself rather than from the hierarchical fusion strategy. A direct fusion comparison would clarify the benefits of the selective approach.

2) Do you consider other structure encoders besides R-GCN, there are multiple encoding models available for protein structure.

3) How to explain the performance differences in FPR across models in Table 2? The substantial FPR variance raises questions about the stability and consistency of SAGEPhos compared to other baselines. A more detailed discussion of the factors behind these differences would be helpful.

---

> ### Author Response · Authors · 2024-11-23
> **To reviewer-1xkZ**
>
> Dear reviewer 1xkZ,
>
> Thank you for raising these valuable points. Our responses are as follows:
>
> `1.Do you consider directly comparing SAGEPhos with a model that simply combines sequence and structure data? It seems the improvements in SAGEPhos may result more from the utilization of structural information itself rather than from the hierarchical fusion strategy. A direct fusion comparison would clarify the benefits of the selective approach.`
>
> We have indeed conducted systematic comparisons between SAGEPhos and baseline models with different feature combination approaches. As presented in our ablation study (Section 4.4, Table 3), we evaluated three distinct fusion strategies:
>
> (1) Direct concatenation of substrate sequence, substrate structure and kinase sequence features (w/o fusion);
>
> (2) Direct concatenation of substrate sequence and structure, then use inter-modal fusion between substrate feature and kinase sequence (w/- inter fusion);
>
> (3) Our proposed hierarchical fusion approach in SAGEPhos (w/- fusion & w/- empha).
>
> For your convenience, we have extracted the relevant experimental results for these three strategies:
>
> | Method | Accuracy↑ | AUC-ROC↑ | AUC-PRC↑ | FPR↓ |
> | :-- | :--: | :--: | :--: | :--: |
> |w/o fusion|65.8±0.7|73.1±0.8| 73.0±0.4 | 41.0±0.8 |
> |w/- inter fusion|77.6±0.2|86.0±0.2|83.8±0.4|27.9±1.4|
> |w/- fusion & w/- empha|80.6±0.2|88.3±0.1|86.2±0.2|21.7±0.4|
>
> The experimental results reveal that the simple concatenation baseline (w/o fusion) achieves notably lower performance across all metrics compared to SAGEPhos (e.g., Accuracy: 65.8% vs 80.6%). Even using inter-modal fusion strategy only (w/- inter fusion) shows limited improvement. This performance gap suggests that our Bio-Augmented fusion strategy (fusion strategy between substrate sequence and substrate structure) not only effectively leverages structural information but also successfully filters noise while preserving essential structural features. These results demonstrate that both the incorporation of structural information and our specialized fusion mechanism contribute synergistically to the model's optimal performance.
>
> `2.Do you consider other structure encoders besides R-GCN, there are multiple encoding models available for protein structure.`
>
> We selected R-GCN as our graph encoder based on its unique ability to model diverse residue-residue relationships in phosphorylation site prediction. Through its relation-specific weight matrices, R-GCN can capture and learn the distinct importance of different types of residue relationships (sequential, spatial, and k-nearest neighbor connections).
>
> To validate this choice, we conducted comprehensive comparisons with representative graph encoders:
>
> | Model| Acc↑| AUC-ROC↑ | AUC-PRC↑ | FPR↓ |
> |--------|------|----------|-----------|------|
> | GCN| 78.2 | 86.1| 84.3| 25.5 |
> | GIN| 78.6 | 85.9| 84.2| 23.7 |
> | GAT| 77.6 | 86.0| 84.2| 27.5 |
> | R-GCN| 80.6 | 88.3| 86.2| 21.7 |
>
> R-GCN consistently outperforms alternatives across all metrics, with notable improvements in accuracy (+2% compared to GIN), AUC-ROC (+2.2% compared to GCN), and a significant reduction in false positive rate (21.7% vs ≥23.7%). While GCN, GIN, and GAT are effective in many graph learning tasks, their uniform treatment of residue relationships limits their ability to capture the nuanced structural information critical for phosphorylation site prediction.
>
> `3.How to explain the performance differences in FPR across models in Table 2? The substantial FPR variance raises questions about the stability and consistency of SAGEPhos compared to other baselines. A more detailed discussion of the factors behind these differences would be helpful.`
>
> SAGEPhos, Phosformer, and Phosformer-ST are designed to predict phosphorylation by considering both kinase and substrate information, while other baselines only utilize substrate information for prediction.
>
> The FPR variations can be primarily attributed to the incorporation of kinase information in SAGEPhos and Phosformer-ST. Compared to substrate sequences, kinase information is relatively sparse, leading to performance fluctuations, including FPR, particularly in kinase cold-start scenarios. This effect is evident even in Phosformer-ST despite its high FPR values. While Phosformer also considers kinase information, its strong bias toward negative predictions results in extremely low FPR with minimal variation, compromising its predictive value.
>
> As for the substrate-only baselines (MusiteDeep, DeepPhos, PhosIDN, PhosIDNSeq), only MusiteDeep and PhosIDN series show relative stability since they only consider substrate information, which is more abundant and thus less affected by cold-start partitioning. However, their higher FPR values, especially PhosIDN, indicate inferior performance. Based on this analysis, the observed FPR variations are natural fluctuations caused by kinase absence, and SAGEPhos demonstrates reliable stability and consistency in its overall performance.

---

> > ### Author Response · Authors · 2024-11-24
> >
> > Dear reviewer 1xkZ,
> >
> > Thank you again for patiently reading our response! We hope we could address your concerns and we are looking forward to your reply for further discussions!

---

> > > ### Comment · Reviewer_1xkZ · 2024-11-27
> > >
> > > Thank you for your response, my concern has been addressed. I also believe that future derivative works based on large models such as AlphaFold or ESM are likely to have a greater impact. Considering all of this, I have decided to raise my score for this paper.

---

> > > > ### Author Response · Authors · 2024-11-27
> > > >
> > > > Dear reviewer 1xkZ,
> > > >
> > > > We sincerely appreciate your positive feedback and are delighted that our revisions have addressed your concerns.
> > > >
> > > > Thank you again for your valuable guidance throughout this review process.

---

> > > > ### Author Response · Authors · 2024-12-03
> > > > **Appreciation to Reviewer 1xkZ**
> > > >
> > > > Dear Reviewer 1xkZ,
> > > >
> > > > We sincerely appreciate your insightful comments and thoughtful suggestions regarding hierarchical fusion strategy and structural encoder, which have helped us significantly improve the quality of our manuscript. We look forward to advancing the development of AI for post-translational modification analysis through our work and hope to provide inspiration for other researchers in this field, and we remain committed to advancing research in this important field.
> > > >
> > > > We are delighted that our revisions have successfully addressed all your concerns and are deeply honored that you have raised the score for our manuscript. Given the highly competitive nature of the current score distribution, we would be immensely grateful if you could consider further increasing the score.
> > > >
> > > > Thank you again for your valuable guidance throughout this review process.

---

### Official Review · Reviewer_dmnR · 2024-11-02

**Soundness:** 3
**Presentation:** 3
**Contribution:** 3
**Rating:** 6
**Confidence:** 2

**Summary:**

The paper discusses SAGEPhos, a model developed for predicting phosphorylation sites in proteins by using both kinase sequence information and the 3D structural information of substrates. It employs a multi-modal fusion framework called MERGE, which integrates structural and sequence data through two main techniques: Bio-Coupled Fusion and Bio-Augmented Fusion. This approach aims to highlight critical phosphorylation regions by combining sequence-based and structure-based insights. The model demonstrates enhanced accuracy and AUC metrics compared to traditional methods, showing potential applications in kinase signaling pathways relevant to diseases such as cancer and neurodegenerative disorders.

**Strengths:**

1. Innovative Fusion Strategy: The paper introduces Bio-Coupled and Bio-Augmented fusion mechanisms, which enhance phosphorylation site prediction by effectively merging structural and sequence data.
2. Integration of Structural Data: Utilizing structural data (such as from AlphaFold2) to predict phosphorylation sites addresses a notable gap in traditional models that rely solely on sequence information, potentially providing more biologically relevant insights.
3. Zero-shot prediction: SAGEPhos is evaluated in different contexts, which test its ability to generalize to unseen kinases, demonstrating versatility.

**Weaknesses:**

1. Limited Justification for Method Choices: The choice of certain architectural elements, like GCN for structural data and gated/residual features, lacks thorough justification. Without clear reasoning, it’s difficult to determine if these choices genuinely improve the model’s performance or if simpler alternatives could suffice.
2. Complex Model Architecture: The use of multi-modal fusion (Bio-Coupled and Bio-Augmented), along with gated and residual features, adds considerable complexity. This complexity may reduce the model’s accessibility and reproducibility for researchers without advanced computational resources or expertise in deep learning.
3. Dependence on AlphaFold Structures: The model relies heavily on AlphaFold’s structural predictions, which may not be entirely accurate for all proteins, especially those without known homologs.

**Questions:**

1. What steps did you take to ensure the model’s generalizability, given that it was only evaluated on a limited set of datasets?
2. Could you clarify why you chose complex models like R-GCN for structural representation over simpler alternatives, and what benefits they provide?
3. How does SAGEPhos handle incomplete or low-confidence structural data, given its reliance on AlphaFold2 predictions?
4. Why does the model only use both sequence and structure data for substrates, while only sequence data is used for kinases?
5. Why was a GCN applied to the AlphaFold structures instead of directly using AlphaFold’s embeddings?

---

> ### Author Response · Authors · 2024-11-23
> **To reviewer-dmnR (1)**
>
> Dear reviewer dmnR,
>
> Thank you for raising these valuable points. Our responses are as follows:
>
> `1.What steps did you take to ensure the model’s generalizability, given that it was only evaluated on a limited set of datasets?`
>
> As reported in our manuscript Appendix C, Table A1, we validated our model's generalizability on CDK17, a kinase absent from our training set, using substrate data from an independent dataset. We have now extended this evaluation to include four additional novel kinases (MYLK4, CDKL1, PHKG2, SRPK3) [1]. Comparing with MusiteDeep, the best-performing baseline, our model demonstrates strong generalization capability across different kinases:
>
> | Method ||SAGEPhos|||MusiteDeep||
> | :----- | :--: | :------: | :------: | :--: | :--------: | :------: |
> || Acc↑|AUC-ROC↑|AUC-PRC↑|Acc↑|AUC-ROC↑|AUC-PRC↑|
> | MYLK4 |84.4|97.9|98.1|71.1|94.5|23.8|
> | CDKL1 |82.6|90.0|91.0|71.1|91.1|4.0|
> | PHKG2|88.2|95.8|95.3|71.8|90.3|15.2|
> | SRPK3|81.3|88.9|88.2|70.6|95.6|8.7|
>
> [1] Johnson et al., 2023, An atlas of substrate specificities for the human serine/threonine kinome.
>
> `2.Could you clarify why you chose complex models like R-GCN for structural representation over simpler alternatives, and what benefits they provide?`
>
> Our choice of R-GCN is motivated by the complex nature of phosphorylation site prediction. Unlike simpler graph models, R-GCN uses relation-specific transformation matrices to effectively capture and learn the distinct importance of different types of residue relationships (sequential, spatial, and k-nearest neighbor connections) in phosphorylation site prediction. Comprehensive comparisons with representative graph models (GCN, GIN, and GAT) demonstrate R-GCN's superior performance across all evaluation metrics:
>
> | Model| Acc↑| AUC-ROC↑ | AUC-PRC↑ | FPR↓ |
> |--------|------|----------|-----------|------|
> | GCN| 78.2 | 86.1| 84.3| 25.5 |
> | GIN| 78.6 | 85.9| 84.2| 23.7 |
> | GAT| 77.6 | 86.0| 84.2| 27.5 |
> | R-GCN| 80.6 | 88.3| 86.2| 21.7 |
>
> `3.How does SAGEPhos handle incomplete or low-confidence structural data, given its reliance on AlphaFold2 predictions?`
>
> To handle varying confidence levels in AlphaFold2's structural predictions, we propose a Dynamic fusiOn meChanism (DOC) that employs a gated module to selectively filter structural information based on relevance. Instead of using hard thresholds to discard low-confidence structures, DOC preserves valuable features while reducing the impact of noisy predictions. Comparative experiments with commonly used pLDDT thresholds (0.5 and 0.7) demonstrate SAGEPhos's superior performance across all metrics:
>
> | **Model** | Acc↑ | AUC-ROC↑| AUC-PRC↑| FPR↓|
> | :-------- | :--: | :------: | :------: | :--: |
> | plDDT>0.7|79.9|87.7|85.8|22.4|
> | plDDT>0.5|80.0|88.0|86.3|22.3|
> | SAGEPhos|80.6|88.3|86.2|21.7|
>
> `4.Why does the model only use both sequence and structure data for substrates, while only sequence data is used for kinases?`
>
> We employ different input features for kinases and substrates based on their distinct biological properties. As shown in previous studies [1], kinases are highly conserved enzymes with similar core structures and functional sites, making their sequence information sufficient for prediction purposes. Furthermore, not all kinase sequences have corresponding structural information available - incorporating both modalities would significantly reduce our training dataset size (from 29,376 to 24,893 samples), which consequently leads to decreased model performance, as shown below:
>
> | **Model** | Data Size | Acc↑ | AUC-ROC↑| AUC-PRC↑| FPR↓|
> | :-------- | :-------: | :--: | :------: | :------: | :--: |
> | add_kin_structure|24,893|78.3|86.2|84.9|23.9|
> | SAGEPhos|29,376|80.6|88.3|86.2|21.7|
>
> In contrast, the high sequence and structural diversity of substrates, which critically influences phosphorylation events, necessitates the integration of both features for accurate prediction.
>
> [1] Taylor et al., 2011, Protein kinases: evolution of dynamic regulatory proteins.
>
> `5.Why was a GCN applied to the AlphaFold structures instead of directly using AlphaFold’s embeddings?`
>
> Given the resource requirements of AlphaFold, we took a more lightweight approach by utilizing pre-computed structures from the AlphaFold Database and processing them with R-GCN. The R-GCN's relation-specific architecture enables explicit modeling of various molecular interactions, which is essential for capturing diverse structural patterns in kinase-substrate interactions.
>
> Specifically, due to resource constraints, AlphaFold2's intensive computational requirements make it impractical to directly generate embeddings for our dataset of 30,000+ sequences, especially within the limited rebuttal period. Our current approach is significantly more efficient while still effectively capturing structural information. Thank you for suggesting this potential direction, we plan to explore it in future work when sufficient computational resources become available.

---

> > ### Author Response · Authors · 2024-11-23
> > **To reviewer-dmnR (2)**
> >
> > Dear reviewer dmnR,
> >
> > Regarding Question 1 about model generalizability, our detailed analysis for the experimental results is as follows:
> >
> > `1.What steps did you take to ensure the model’s generalizability, given that it was only evaluated on a limited set of datasets?`
> >
> > For your convenience, we extract the relevant experimental results from Comment (1):
> >
> > | Method ||SAGEPhos|||MusiteDeep||
> > | :----- | :--: | :------: | :------: | :--: | :--------: | :------: |
> > || Acc↑|AUC-ROC↑|AUC-PRC↑|Acc↑|AUC-ROC↑|AUC-PRC↑|
> > | MYLK4 |84.4|97.9|98.1|71.1|94.5|23.8|
> > | CDKL1 |82.6|90.0|91.0|71.1|91.1|4.0|
> > | PHKG2|88.2|95.8|95.3|71.8|90.3|15.2|
> > | SRPK3|81.3|88.9|88.2|70.6|95.6|8.7|
> >
> > To understand these performance differences, we analyze the underlying methodology of MusiteDeep. MusiteDeep employs an 11-size sliding window approach, marking center amino acids as 1 (phosphorylation site) or 0 (non-site). The natural rarity of phosphorylation sites leads to significant class imbalance, which explains MusiteDeep's distinct performance pattern: high AUROC but low AUPRC. This pattern emerges because AUROC remains stable due to FPR's consideration of total negative samples (FPR = FP/(FP+TN)), while AUPRC is more sensitive to imbalance as precision (TP/(TP+FP)) directly compares false positives to true positives without accounting for the total negative sample size.
> >
> > The impact of this methodological limitation becomes evident in real-world scenarios. While MusiteDeep achieves comparable AUC-ROC scores, its significantly lower AUC-PRC values (4.0-23.8%) indicate poor performance in identifying true phosphorylation sites under highly imbalanced conditions. In contrast, SAGEPhos demonstrates robust generalization ability by maintaining superior performance across all metrics (Accuracy: 81.3-88.2%, AUC-ROC: 88.9-97.9%, AUC-PRC: 88.2-98.1%), particularly in precisely detecting phosphorylation sites in real-world applications.

---

> > > ### Author Response · Authors · 2024-11-24
> > >
> > > Dear reviewer dmnR,
> > >
> > > Thank you again for patiently reading our response! We hope we could address your concerns and we are looking forward to your reply for further discussions!

---

> > > > ### Comment · Reviewer_dmnR · 2024-11-26
> > > >
> > > > Thank you for response. The new version resolved all my concerns and I recommend for accepting this paper.

---

> > > > > ### Author Response · Authors · 2024-12-03
> > > > > **Appreciation to Reviewer dmnR**
> > > > >
> > > > > Dear Reviewer dmnR,
> > > > >
> > > > > We sincerely appreciate your insightful comments and thoughtful suggestions regarding model generalization and structural information incorporation, which have helped us significantly improve the quality of our manuscript. We look forward to advancing the development of AI for post-translational modification analysis through our work and hope to provide inspiration for other researchers in this field, and we remain committed to advancing research in this important field.
> > > > >
> > > > > We are truly grateful that you have championed our paper and are pleased that our revisions have successfully addressed all your concerns. Given the highly competitive nature of the current score distribution, we would be deeply appreciative if you could consider increasing the score.
> > > > >
> > > > > Thank you again for your valuable guidance throughout this review process.

---

> ### Author Response · Authors · 2024-11-27
>
> Dear reviewer dmnR,
>
> We sincerely appreciate your positive feedback and are delighted that our revisions have addressed your concerns. We believe our work brings novel perspectives to post-translational modification analysis, particularly in phosphorylation studies, and we are committed to continuing our research endeavors in this important field. If possible, we would be grateful if you could consider increasing the score.
>
> Thank you again for your valuable guidance throughout this review process.

---

### Official Review · Reviewer_aM45 · 2024-11-04

**Soundness:** 3
**Presentation:** 3
**Contribution:** 3
**Rating:** 6
**Confidence:** 4

**Summary:**

This work aims at phosphorylation site prediction. While the problem of detecting phosphorylation sites is possible through high-throughput experimental techniques such as mass spectrometry, predicting phosphosites for a speicific kinase is still a relevant computational problem as detecting kinase-substrate associations is more difficult.  The work presents SAGEPhos. The contribution of this method is that as opposed to traditional methods that rely solely on sequence information, SAGEPhos  leverages both kinase sequence and structural information. SAGEPhos combines these modalities through gated fusion strategies. The inter-fusion and intra-fusion strategies aim at capturing information on different modalities for both kinase and the substrate. The paper reports a 12% improvement in AUC-ROC compared to other methods.

**Strengths:**

Strengths:

- The fusion strategies and the gated architecture allow for combining information such as structural or conservation scores and sequence information in a useful way.
- There is an ablation study to understand the impact of inter and intra-fusion models.
- The authors compare their methods with the most relevant work.
- The authors conduct experiments to assess the sensitivity to hyperparameter changes.
- Authors conduct experiments to understand specific cases. The case studies on GSK3B and MK01 are interesting. This work could be extended to other kinases to understand  the kinase specific motifs.

**Weaknesses:**

- The main weakness of the paper is in the presentation. Many critical experimental details lack clarity which I listed in the questions:
-  The main contribution of the paper is adding structural information to the model. A comparison to a structure-aware protein language is missing. For example, how would SaProt perform on this task?
The paper claims zero-shot generalization but only demonstrates this on a single kinase. The zero-shot experiments could be more comprehensive. Additionally, a zero-shot classifier for kinases, DeepKinZero (Deznabi et al. 2020), exists, but the authors did not discuss its relevance to their work.
Authors should also discuss in the report how performance changes across different kinases. Are certain kinases easier to predict than others? Is the kinase-specific performance related to the number of substrates associated with each kinase?

- The discussion section could be strengthened. For example, MusiteDeep's performance in cold-start cases, achieving only 6.4% AUPRC or 0.8% AUPRC, should be mentioned. Also, the authors use AlphaFold structures. Using PDB structures whenever available would make more sense, filling in only missing ones with AlphaFold predictions. Authors should at least acknowledge this and discuss the potential biases it could lead to.

**Questions:**

- It is unclear how the cold-start and warm-start experiments are designed. The cited reference does not include this information either, which is critical to assessing the results.

- The negative example selection procedure is not clearly described. The authors state: “For each kinase, we selected an equal number of negative samples from substrate sequences lacking explicit evidence of catalysis by that kinase, resulting in a balanced 1:1 ratio of positive to negative “ Are negative sequences chosen from other positions within the same substrate of the kinase, or the entire substrate set across all kinases? Are these positions known phosphosite locations on other substrates, or are they potential sites that could accept a phosphate group without a reported kinase association? Clarifying this distinction is important, as each case poses a different level of classification difficulty.

- The paper combines three different datasets, removing redundant information. When splitting the train/test folds, is substrate similarity or kinase similarity taken into account?

Some other  technical details are missing, which will make it difficult to reproduce the work:


-ESM2 is used for feature extraction, but the specific ESM2 model size is not specified.
- How are the conservation scores computed?
- Which physicochemical properties are used and with what kind of representation? Continuous scale, binarized categories? Figure A1 needs a legend for the color scale.
“We used this data to create a new test set, where substrate-kinase pairs ranked first for CDK17 were treated as positive samples, and those ranked the bottom two were designated as negative samples.” How many substrates are taken from the bottom and the top?

---

> ### Author Response · Authors · 2024-11-23
> **To reviewer-aM45**
>
> Dear reviewer aM45,
>
> Thank you for raising these valuable points. Our responses are as follows:
>
> `1.It is unclear how the cold-start and warm-start experiments are designed. The cited reference does not include this information either.`
>
> For warm-start setting, the dataset was randomly split into training, validation, and testing sets with a ratio of 8:1:1;
>
> For kinase cold-start, data points with the same kinase were grouped into the same set to prevent kinase overlap across train/valid/test sets;
>
> For substrate cold-start, data points with identical substrate sequences were grouped together to ensure no sequence overlap between sets.
>
> These experimental designs evaluate model generalization across random, unseen kinase, and unseen substrate scenarios.
>
> `2.Are negative sequences chosen from other positions within the same substrate of the kinase, or the entire substrate set across all kinases? Are these positions known phosphosite locations on other substrates, or are they potential sites that could accept a phosphate group without a reported kinase association? `
>
> Our negative sequences are drawn from the entire substrate set across all kinases, comprising two types:
>
> (1) Known phosphosites that are validated substrates of other kinases;
>
> (2) Experimentally verified phosphorylatable sites without reported kinase associations (majority)
>
> For each kinase, we maintained a 1:1 ratio between positive examples (confirmed phosphorylation sequences) and randomly selected negative examples from these two sources. We have added detailed description of this negative example selection procedure to Appendix A "DATASETS AND PARTITION METHOD".
>
> `3.When splitting the train/test folds, is substrate similarity or kinase similarity taken into account?`
>
> Rather than using sequence similarity thresholds, we enforce strict data separation by grouping identical kinases or substrates into the same split named cold-start. To test model generalization, we conducted additional experiments using sequence similarity-based splits with MMseqs2 (50% similarity threshold, 8:1:1 ratio) for both kinase sequences and substrate sequences respectively.
>
> Our results demonstrate that sequence similarity-based clustering achieves relatively better performance (kinase clustering: Acc 78.8%, AUC-ROC 86.4%; substrate clustering: Acc 79.2%, AUC-ROC 87.3%) compared to the more challenging cold-start settings (kinase cold-start: Acc 68.6%, AUC-ROC 76.9%; substrate cold-start: Acc 79.1%, AUC-ROC 87.2%). This performance gap validates that our cold-start splitting strategy poses a more rigorous evaluation scenario than sequence similarity-based splits. As expected, these results are lower than those obtained under random split settings (Acc: 80.6%, AUC-ROC: 88.3%), which aligns with the intuition that performance naturally decreases as the splitting criteria become more stringent.
>
> `4.ESM2 is used for feature extraction, but the specific ESM2 model size is not specified.`
>
> We used ESM-2-650M for feature extraction in our experiments. This has been added to Appendix B “IMPLEMENTATION DETAILS” in revised manuscript.
>
> `5.How are the conservation scores computed?`
>
> The conservation scores in our method are implemented as learnable embeddings specifically for phosphorylation sites. This design is motivated by the biological observation that different positions in protein sequences exhibit varying degrees of conservation, with functionally critical sites typically being more conserved. In particular, the central residue (S/Y/T) within the 11-mer peptide sequences represents a highly conserved motif for phosphorylation events. These scores are dynamically optimized during model training to capture the relative importance of these conserved sites.
>
> The effectiveness of this approach is demonstrated in Section 4.4 "ABLATION STUDY", where we compare model performance with and without the conservation score emphasis ("w/- fusion & w/- empha" vs. "w/- fusion & w/o empha").
>
> `6.Which physicochemical properties are used and with what kind of representation? Continuous scale, binarized categories?`
>
> We employ four fundamental physicochemical properties commonly used in protein structure-function analysis: (1) Aliphatic, (2) Aromatic, (3) Acidic charged, and (4) Basic charged. Each property is represented as a binary category. These basic properties capture essential amino acid characteristics that influence protein structure and interactions. We have added this description to Appendix B "IMPLEMENTATION DETAILS AND HYPERPARAMETERS" in revised manuscript.
>
> `7.Figure A1 needs a legend for the color scale.`
>
> We have added a legend of the color scale in Figure A1.
>
> `8.In CDK17, how many substrates are taken from the bottom and the top?`
>
> After structural matching and balancing the dataset to maintain a 1:1 ratio between positive and negative samples, our final test set consisted of 46 positive samples (ranked first) and 46 negative samples (ranked at the bottom).

---

> > ### Author Response · Authors · 2024-11-24
> >
> > Dear reviewer aM45,
> >
> > Thank you again for patiently reading our response! We hope we could address your concerns and we are looking forward to your reply for further discussions!

---

> > > ### Comment · Reviewer_aM45 · 2024-11-27
> > >
> > > Thank you for answering and uptading the results. Authors partly addresssed my concerns. The presentation is improved because may missing information in the experimental design is added . Based on that I will improve my presentation score.
> > > But my concerns on the evaluation remains:
> > > 1) I think the cold-start kinae or substrate experiments are not truly cold. Some kinases are very similar  for example keeping AKT1 in the training set and putting AKT2 in the test set, the test cases for AKT2 would be easy. Sama for the substrate. Thus the cold start experiments could have taken the similarity of the sequences.
> > >
> > > 2)Authors learn the conservation scores. A natural question to ask is would the pre-calculated conservation scores from the MSAs would better than that. And how is it different than an attention score.
> > >
> > > 3) The comment on "The zero-shot experiments could have been more comprehensive. " is not handled.

---

> > > > ### Author Response · Authors · 2024-12-01
> > > > **To reviewer-aM45 (1)**
> > > >
> > > > Dear reviewer aM45,
> > > >
> > > > We are deeply grateful for your thoughtful feedback. Please find our detailed responses as follows:
> > > >
> > > > `1. I think the cold-start kinase or substrate experiments are not truly cold. Some kinases are very similar for example keeping AKT1 in the training set and putting AKT2 in the test set, the test cases for AKT2 would be easy. Sama for the substrate. Thus the cold start experiments could have taken the similarity of the sequences. `
> > > >
> > > > Although it's true that similar sequences could potentially be split between training and test sets, when we analyzed the sequence similarity between training and test sets using a threshold of 0.5, we found that only a tiny fraction of test sequences showed high similarity with the training data.
> > > >
> > > > Specifically:
> > > >
> > > > （1）In the cold_kinase test set (3522 kinase sequences), only 6 sequences (0.17%) showed similarity above 0.5 with training sequences,
> > > >
> > > > （2）In the cold_substrate test set (3612 substrate sequences), no sequences exceeded the 0.5 similarity threshold with training sequences.
> > > >
> > > > These statistics indicate that our cold-start setting effectively maintains sequence diversity between training and test sets, as 99.83% of test sequences in cold_kinase and 100% in cold_substrate show low similarity (< 0.5) with the training set. To rigorously evaluate the potential influence of the 6 identified high-similarity sequences, we conducted a comparative analysis by removing these sequences from the cold_kinase test set. We compared the model performance between the original test set ('cold_kinase') and the filtered test set without high-similarity sequences ('cold_kinase_sim'):
> > > >
> > > > | **Model** | Acc↑| AUC-ROC↑| AUC-PRC↑| FPR↓|
> > > > | :-------- | :--: | :------: | :------: | :--: |
> > > > |cold_kinase_sim|68.5|76.9|76.0|22.6|
> > > > |cold_kinase|68.6|76.9|75.9|22.4|
> > > >
> > > > The results show almost no difference in model performance, which confirms our original test setup was reliable.
> > > >
> > > > `2. Authors learn the conservation scores. A natural question to ask is would the pre-calculated conservation scores from the MSAs would better than that. `
> > > >
> > > > We opt for learnable embeddings to derive conservation scores rather than pre-calculated MSA-based conservation scores because our approach learns task-specific conservation patterns during training while leveraging ESM's pre-trained representations, enabling the model to capture phosphorylation-related features that static MSA-based methods might overlook.
> > > >
> > > > And we appreciate your suggestion to calculate conservation scores through MSA searches, and we attempted to explore this option. But when we tried to conduct BLAST searches on several databases (NR, UniProt_TrEMBL, and SwissProt), we encountered practical limitations in each case. Specifically:
> > > >
> > > > (1) For large-scale databases like NR and UniProt_TrEMBL, the computational cost is prohibitive - searching our dataset of over 30,000 sequences would require more than 15 days of continuous computation, given that each 11-residue query takes 45 seconds to 3 minutes.
> > > >
> > > > (2) While SwissProt searches are computationally feasible, they provide insufficient sequence hits (average <5 hits per query) for reliable conservation analysis. This leads to statistically unreliable conservation scores with artificially high minimum values (>0.2) and limited discriminative power due to the compressed score range.
> > > >
> > > > Based on these considerations, we choose the learnable embedding-based approach which provides a lightweight method for dynamic learning of phosphorylation-related features compared to MSA-based methods. We will compare our approach with MSA-based conservation score calculations when time and computational resources permit.
> > > >
> > > > `3. And how is it different than an attention score. `
> > > >
> > > > The key difference between these two scoring mechanisms lies in their scope and biological relevance. While attention scores capture global sequence-wide relationships between different positions, conservation scores incorporate valuable prior knowledge by focusing on conserved positions that reflect local, position-specific features. To demonstrate the effectiveness of these different approaches, we designed experiments for comparison, where we calculated attention scores using self-attention mechanism:
> > > >
> > > > |Model|Acc↑|AUC-ROC↑ |AUC-PRC↑ |FPR↓ |
> > > > |--------|------|----------|-----------|------|
> > > > |attention score|77.8|85.9|84.1|25.7|
> > > > |SAGEPhos|80.6|88.3|86.2|21.7|
> > > >
> > > > The experimental results show that SAGEPhos consistently outperforms the attention score method, demonstrating the necessity of incorporating domain-specific prior knowledge through conservation scores.

---

> > > > > ### Author Response · Authors · 2024-12-01
> > > > > **To reviewer-aM45 (2)**
> > > > >
> > > > > `4. The comment on "The zero-shot experiments could have been more comprehensive. " is not handled. `
> > > > >
> > > > > We appreciate your concern about the comprehensiveness of zero-shot experiments. As initially reported in our manuscript's Appendix C, Table A1, we validated our model's generalizability using CDK17, a kinase absent from our training set. To address the comprehensiveness issue, we have now substantially expanded our zero-shot evaluation to include four additional novel kinases (MYLK4, CDKL1, PHKG2, SRPK3) from as same novel dataset as CDK17[1]. We computed sequence similarities between each zero-shot kinase and all kinases in our training set and recorded the highest similarity. The comparative analysis with MusiteDeep, our strongest baseline, demonstrates SAGEPhos's robust generalization capability:
> > > > >
> > > > > |Method|Highest_sim||SAGEPhos|||MusiteDeep||
> > > > > | :----- | :------: | :--: | :------: | :------: | :--: | :--------: | :------: |
> > > > > |||Acc↑|AUC-ROC↑|AUC-PRC↑|Acc↑|AUC-ROC↑|AUC-PRC↑|
> > > > > |CDK17|0.328|91.6|97.4|97.6|69.3|94.2|4.8|
> > > > > |MYLK4|0.163|84.4|97.9|98.1|71.1|94.5|23.8|
> > > > > |CDKL1|0.142|82.6|90.0|91.0|71.1|91.1|4.0|
> > > > > |PHKG2|0.726|88.2|95.8|95.3|71.8|90.3|15.2|
> > > > > |SRPK3|0.288|81.3|88.9|88.2|70.6|95.6|8.7|
> > > > >
> > > > > These results, spanning diverse kinase families with predominantly low sequence similarities to the training set (4 out of 5 kinases showing similarity scores <0.4), demonstrate strong generalization capability, especially for kinases like MYLK4 and CDKL1 with very low similarity scores. While MusiteDeep achieves similar AUC-ROC scores, its AUC-PRC values are 4.0-23.8% lower, indicating poor detection of true phosphorylation sites. This limitation arises from its conversion of sequence data to highly imbalanced site-level samples. In contrast, SAGEPhos demonstrates robust generalization ability by maintaining superior performance across all metrics (Accuracy: 81.3-91.6%, AUC-ROC: 88.9-97.9%, AUC-PRC: 88.2-98.1%), highlighting its effectiveness in handling diverse kinase substrates.
> > > > >
> > > > > Regarding another zero-shot kinase predictor, DeepKinZero [2], we acknowledge its significant contribution but faced challenges in direct comparison. While our study covers over 700 kinase types, DeepKinZero focuses on 458 common kinases. Additionally, the unavailability of their Kinase Feature generation code prevented us from extending the analysis to our broader kinase set. We appreciate your suggestion and will include a discussion of DeepKinZero in our revised manuscript, acknowledging its contributions and explaining these constraints.
> > > > >
> > > > > [1] Johnson et al., 2023, An atlas of substrate specificities for the human serine/threonine kinome.
> > > > >
> > > > > [2] Deznabi et al., 2020, DeepKinZero: zero-shot learning for predicting kinase–phosphosite associations involving understudied kinases.

---

> > > > > > ### Author Response · Authors · 2024-12-01
> > > > > >
> > > > > > Dear reviewer aM45,
> > > > > >
> > > > > > We are deeply grateful for your constructive feedback and insightful suggestions, which have significantly helped us improve our work. We sincerely hope our responses have adequately addressed your questions and concerns. We will carefully incorporate all these analyses and discussions into our camera-ready version. If you have any additional questions or concerns, we would be honored to engage in further discussions. Thank you again for your invaluable guidance throughout this review process.

---

> > > > > ### Comment · Reviewer_aM45 · 2024-12-01
> > > > > **conservation scores**
> > > > >
> > > > > Thank you for answering my questions.
> > > > > 1) What was the similarity metric? Identitiy? And for the kinases, the similarity should be calculated over the kinase domains I think. Otherwise the unrelated domains in the protein would reduce the similarity.
> > > > >
> > > > > 2) One last question, what gurantees that the learned scores reflect conservation scores?  I think the authors should showcase that the learned scores are in the conservation scores in the evolution - which is the  common term that the field has been using.

---

> > > > > > ### Author Response · Authors · 2024-12-03
> > > > > >
> > > > > > Dear reviewer aM45,
> > > > > >
> > > > > > We are deeply grateful for your thoughtful feedback. Please find our detailed responses as follows:
> > > > > >
> > > > > > `1. What was the similarity metric? Identity? And for the kinases, the similarity should be calculated over the kinase domains I think. Otherwise the unrelated domains in the protein would reduce the similarity.`
> > > > > >
> > > > > > For the similarity metric, we used the Jaccard similarity (also known as Tanimoto similarity) calculated based on k-mers of protein sequences, rather than sequence identity. This metric measures the overlap between sequence fragments while accounting for their compositional similarities.
> > > > > >
> > > > > > However, we fully agree with your suggestion about focusing on kinase domains to avoid interference from unrelated protein regions. Following your recommendation, we have used kinase domains to calculate kinase similarity. First, we extracted kinase domains using HMMER with Pfam domain definitions. Subsequently, we employed BLAST (blastp) to calculate sequence similarities between these extracted domains, as it better accounts for sequence identity.
> > > > > >
> > > > > > We found that when calculating similarity on kinase domains, there exists considerable inherent similarity among kinases, which is expected given their conserved catalytic functions. This observation aligns with previous studies showing high conservation of kinase domains across different kinases [1]. In our cold_kinase dataset, out of 3,516 test sequences, 840 showed less than 50% similarity to the training set sequences based on this kinase domain similarity calculation method.
> > > > > >
> > > > > > To further validate SAGEPhos's generalization capability, we tested these 840 low-similarity samples against MusiteDeep. The experimental results are as follows:
> > > > > >
> > > > > > |Model|Acc↑|AUC-ROC↑|AUC-PRC↑|FPR↓|
> > > > > > |--------|------|----------|-----------|------|
> > > > > > |SAGEPhos|57.4|60.3|61.7|38.1|
> > > > > > |MusiteDeep|47.4|35.8|2.8|51.9|
> > > > > >
> > > > > > The results demonstrate that even when considering kinase domain similarity, SAGEPhos still comprehensively outperforms our strongest baseline, MusiteDeep. We believe that your suggestion provides a more stringent and accurate method for calculating enzyme similarity, and indeed all algorithms show some performance decrease under these stricter conditions. In our final version, we will present and analyze experimental results using different enzyme similarity calculation methods (Jaccard similarity, MMSeq2 clustering, and kinase domain similarity).
> > > > > >
> > > > > > [1] Taylor et al., 2011, Protein kinases: evolution of dynamic regulatory proteins.
> > > > > >
> > > > > > `2. One last question, what guarantees that the learned scores reflect conservation scores? I think the authors should showcase that the learned scores are in the conservation scores in the evolution - which is the common term that the field has been using.`
> > > > > >
> > > > > > Thank you for raising this important point about the relationship between our learned scores and evolutionary conservation. To validate whether our learned scores capture conservation patterns, we conducted a systematic correlation analysis between our approach and traditional MSA-based conservation scores.
> > > > > >
> > > > > > To investigate this relationship, due to time constraints, we selected 100 sequences from our test set and calculated their MSA-based conservation scores using the NR database. Our model learns a weight vector for each central position. We computed the average of these weights to obtain a single score per position, enabling direct comparison with MSA-based conservation scores.
> > > > > >
> > > > > > Our analysis revealed a moderate positive correlation (Spearman correlation coefficient = 0.5135) between these two measures, suggesting that our learned weights partially capture evolutionary conservation signals. However, it is worth noting that the simple averaging of learnable scores may not fully reflect the relationship between these two measures. Moreover, some differences between our task-specific learnable scores and MSA-based conservation scores are reasonable, given their distinct calculation methods.
> > > > > >
> > > > > > To avoid potential confusion, we will revise our terminology from "conservation score" to "learnable score" in the final manuscript to better reflect the nature of our method.

---

> > > > > > ### Author Response · Authors · 2024-12-03
> > > > > > **Appreciation to Reviewer aM45**
> > > > > >
> > > > > > Dear reviewer aM45,
> > > > > >
> > > > > > We sincerely appreciate your insightful comments and thoughtful suggestions, which have helped us significantly improve the quality of our manuscript. We hope our revisions adequately address your concerns. We look forward to advancing the development of AI for post-translational modification analysis through our work and hope to provide inspiration for other researchers in this field, and we remain committed to advancing research in this important field. If our responses have satisfactorily addressed your concerns, we would be grateful if you could consider increasing the score.
> > > > > >
> > > > > > Thank you again for your valuable guidance throughout this review process. Please do not hesitate to contact us if you have any additional questions or concerns.

---

### Meta-Review · Area_Chair_LW8U · 2024-12-21

**Metareview:**

This paper proposes a novel method for phosphorylation site prediction, called SAGEPhos (Structure-aware kinAse-substrate bio-coupled and bio-auGmented nEtwork for Phosphorylation site prediction).
The work introduces a "Bio-Coupled Modal Fusion" scheme that aims to distill kinase sequence information and combines it with "Bio-Augmented Fusion" that incorporates the spatial information from predicted structures into the sequence information.
The reviewers note that the overall approach is novel and interesting and that the proposed fusion strategy can lead to better prediction.
The performance evaluation results and ablation studies also demonstrate the advantages of SAGEPhos.
However, the paper can benefit from providing further rationale/justifications for the proposed model architecture and design choices and providing a more extensive comparison against other methods (esp., structure-aware methods).
Additional experiments and insights regarding sensitivity to structure prediction results and studies incorporating other structure prediction methods would also strengthen the work.

**Additional Comments On Reviewer Discussion:**

The authors have provided additional explanations and experimental results to address the reviewers' initial concerns, which have partly addressed them.
This has enhanced the reviewers' overall confidence but there remains room for further improvement as summarized above.

---

### Decision · Program_Chairs · 2025-01-22

Accept (Poster)